# Humic Acids Formation during Compositing of Plant Remnants in Presence of Calcium Carbonate and Biochar

**Nataliya Orlova [1,\*], Elena Orlova [1], Evgeny Abakumov [2]**, **Kseniia Smirnova [1] and Serafim Chukov [3]**

[1] Department of Agrochemistry, Faculty of Biology, Saint Petersburg State University, 199178 St. Petersburg, Russia
[2] Department of Applied Ecology, Faculty of Biology, Saint Petersburg State University, 199178 St. Petersburg, Russia
[3] Department of Soil Science and Soil Ecology, Institute of Earth Science, Saint Petersburg State University, 199178 St. Petersburg, Russia
[\*] Correspondence: norlova48@mail.ru; Tel.: +7-9219271664

**Abstract:** The investigation of the mechanisms organic matter transformation in compost organic fertilizers is an urgent task of modern soil ecology and soil chemistry. The main components of such fertilizers are newly formed, weakly humified labile humic acids (HAs). The objective of the study is to determine the mechanism of converting the newly formed HAs into the forms with increased resistance to microbiological and biochemical influences. Obtained during the plant residues decomposition, HAs were studied in the incubation experiment (0, 30, 90 days). Calcium carbonate and biochar produced by rapid pyrolysis from birch and aspen wood, at 550 °C, were used as the composting mixture compounds. Decomposed plant residues—fresh aboveground mass of clover (*Trifolium pratense* L.), rye (*Secale cereale* L.), as well as dry oat straw (*Avena sativa* L.) were the material used for humification. To obtain Has, 0.1 M NaOH and 0.1 M $Na_4P_2O_7$ were used. Then, HAs were separated from fulvic acids (FAs) using a 0.5 M $H_2SO_4$. The amount of labile HAs (HAs1) was estimated by their content in 0.1 M NaOH. The amount of stabilized HAs (HAs2) was calculated by the difference between the HAs content in 0.1 M $Na_4P_2O_7$ and 0.1 M NaOH. Preparation of HAs for elemental composition and NMR analysis was performed according to the International Humic Substances Society's recommendations. The possibility of converting newly formed HAs into stable forms (calcium humates), whose share in the HAs composition reaches 40–50%, has been shown. However, the mechanism of HAs transformation under the studied reagent's influence was different. In the presence of calcium carbonate, it is caused by the physicochemical processes of newly formed HAs rearrangement. However, in the presence of biochar, this is due to the humification processes' intensification and to the increase in the aromatization degree confirmed by the increase in the optical density, as well as by the increase in carbon and oxygen proportion, and by the decrease in hydrogen proportion in HAs molecules. The understanding of HAs formation and transformation mechanisms at the early humification stages can help to optimize the methods of obtaining organic fertilizers.

**Keywords:** organic waste; compost; humic substances; humification; organic fertilizers

## 1. Introduction

Rationally organized farming systems include regular replenishment of humus reserves, primarily by the use of organic fertilizers. Currently, most organic fertilizers are represented by composts mainly consisting of newly formed, weakly humified labile compounds, and hard-to-hydrolyze organic residues that have not passed the stage of humification [1–3]. Once in the soil, the former are easily biodegradable, mineralize quickly, and do not reach deep chemical maturity; the latter are poorly transformed and replenish the insoluble humus residue. With the prolonged use of organic fertilizers, the soil humus quality may gradually decrease; the aliphatic part of HAs molecules increases and the degree of aromatic decreases, which causes the HAs structure "loosening" and a decrease

in their resistance to biodegradation [4,5]. The efficiency of fertilizers based on organic waste compost could be improved by attempting to convert compost humic substances into forms more resistant to microbiological and biochemical influences, and the proportion of hard-to-hydrolyze compounds could be reduced by activating humification processes.

In this context, questions about the composting methods, systems optimization, and the humification processes activation arise. In recent years, technologies for the joint composting of waste with mineral or organic additives have been developed to intensify the process and/or improve the quality of the resulting product [6–8]. The most effective recipes for compostable mixtures are elaborating [9–11]. Some reports indicate the use of biochar as a reagent for composting [1,12–14]. Biochar is a promising soil ameliorant and is actively being introduced into modern agricultural technologies in order to increase soil fertility and crop productivity [15,16]. There is not enough information about the effect of biochar on composting. It is shown that the use of biochar as an additive for composting provides an increase in aeration, the decrease in ammonia volatilization, and improvement in the final product [8,17]. An increased abundance of bacteria, fungi, and actinomycetes, was found in compost with biochar compared to compost without it [18,19]. At the same time, there is very little information on how the biochar effects organic matter humification in composting [20]. Accordingly, of great interest are the studies indicating an increase in the stability of biocarbon compost, due to the stabilized humic substances formation [1,21,22]. However, there is practically no information about the mechanisms governing this process.

Humic substances are a complex structure consisting of various groups characterized by unequal resistance to biodegradation. The most mature and bio thermodynamically stable representatives of humic substances are HAs [23,24]. The HAs special role in the accumulation of humus, and ensuring its ecological functions, is due to structural features, the variety of forms, and the specificity of the organic-mineral derivatives properties. At the same time, both labile and stable forms are distinguished in the HAs group [25]. The division of HAs into these two main categories—labile and stable—is carried out depending on their ability to transform under biochemical influence. Labile HAs are characterized by the degree of low chemical maturity and by the large proportion of aliphatic structures. They are biodegraded easily and mineralize quickly. Stable HAs (more correctly referred to as "relatively stable") are characterized by the significant aromatization degree, the pronounced aromatic core, high optical density, and the increased content of carboxyl groups. They are much more difficult to decompose by microorganisms and as a result they persist for a long time. Labile and stable HAs differ significantly in their agronomic value. Labile substances are the main component of compost based on organic residues.

In order to improve the quality of compost, the question arises about the possibility of obtaining labile humic substances in forms relatively resistant to biochemical influences when composting plant residues. In soils, such properties are possessed by HAs with great chemical activity, capable of interactions by ionogenic bond types. They form compounds with alkaline and alkaline earth cations, primarily with Ca and in this form to microbiological influences [26–28]. However, the presence of this fraction in the newly formed compounds is an unusual phenomenon. Even the HAs of podzolic type soils, which are characterized by higher chemical maturity compared to newly formed HAs, have an extremely weak affinity for calcium, and form humates at very high concentrations of this element in the soil solution, for example, during liming [29,30]. Therefore, in order to appear, the possibility of humates formation during composting ought to increase the chemical maturity degree (the humification depth) of newly-formed compounds, to change the process conditions by the increasing concentration of Ca. To realize these conditions, calcium carbonate is supposed to be used as a composting component, as well as biochar, which is characterized by elevated enrichment with mineral nutrition elements including a very high content of Ca. The appearance of this fraction in the compost composition may significantly increase their efficiency, and increase the stabilization of humus when applied to the soil.

Thus, the aim of the study was to determine the possibility of converting some newly formed HAs into the forms with increased resistance to microbiological and biochemical influences. For this purpose, the nature of plant residue mineralization and humification in the presence of calcium carbonate and biochar, as well as the evolution of the newly formed humic acids composition and structure, ought to be examined based on the determination of their optical density, elemental composition, and molecular structure. Thus, on the basis of these studies, a hypothesis will be tested according to which stabilized and resistant forms of HAs can be formed under the calcium carbonate and biochar influence.

## 2. Materials and Methods

### 2.1. Characteristics of Plant Material and Biochar

The material for humification was fresh, aboveground mass of clover (*Trifolium pratense* L.), fresh aboveground mass of rye (*Secale cereale* L.), and dry oat straw (*Avena sativa* L.). The C/N ratio is one of the main indicators directly affecting the organic matter decomposition rate and compost maturation [13,17,31]. The plant residues studied differed in their availability to decomposition and were characterized by different high (clover), medium (rye), and low (oats), nitrogen enrichment (Table 1).

**Table 1.** Concentration of C, N, and ash, in plant material.

| Plant | C | N | Ash | C/N |
|---|---|---|---|---|
| | % | | | |
| Clover | $41.6 \pm 0.8$ | $1.88 \pm 0.02$ | $5.34 \pm 0.01$ | 22.1 |
| Rye | $43.2 \pm 1.6$ | $1.12 \pm 0.05$ | $5.88 \pm 0.03$ | 38.6 |
| Oats | $42.0 \pm 1.0$ | $0.61 \pm 0.08$ | $5.72 \pm 0.04$ | 68.4 |

$\pm$ standard deviation, at $p < 0.05$, $n = 3$.

Chemically pure $CaCO_3$ and biochar obtained by rapid pyrolysis from birch and aspen wood, at a temperature of 550 °C, were used as the tested additives to plant material that activated the plant residue transformation process. Characteristics of biochar: $C_{org}$ = 85.6%; $N_{tot}$ = 0.43%; $pH_{H_2O}$ = 8.1; ash content = 1.8%; particle size 0.5–2 cm. A more detailed description of the biochar was given earlier [32].

### 2.2. Incubation Experiment

The incubation experiment was conducted to study the effect of $CaCO_3$ and biochar on plant material transformation under composting.

Plant material for incubation was fresh, aboveground mass of clover (*T. pratense* L.), fresh, aboveground mass of rye (*S. cereale* L.), and dry oat straw (*A. sativa* L.). The fresh plant leaves were cut into 2 mm pieces. Plant mass was thoroughly mixed (in 1:5) with quartz sand that was previously ignited at 700 °C so as to devoid it of organic matter. The initial organic carbon content ($C_0$) in the plant-sand mixture is noted in Table 2. $CaCO_3$ and biochar were added into the plant-sand mixture and thoroughly mixed with the samples at the start of the experiment. The concentration of $CaCO_3$ was 5% and biochar was 1%. The studied reagents concentrations were made on the basis of our own preliminary experiments and literature data [8,33,34]. The incubation experiment was carried out in 300 mL plastic vessels at a temperature of 25 °C and optimal compost humidity (60% of total water holding capacity) for 90 days. The repetition of the experience was 4-fold. Dynamic sampling was performed at 30 and 90 days.

**Table 2.** Effect of calcium carbonate and biochar on the carbon content of decomposing plant residues.

| Variants | | Initial Content | 30 Days of Composting | | | 90 Days of Composting | | |
|---|---|---|---|---|---|---|---|---|
| | | $C_0$, % on Dry Weight, | C, % on Dry Weight | C, % on Initial Content | C, % of the Content in the Control | C, % on Dry Weight | C, % on Initial Content | C, % of the Content in the Control |
| Clover | control | 6.94 ± 0.08 | 3.34 ± 0.04 | 48.1 | - | 2.88 ± 0.06 | 41.5 | - |
| | $CaCO_3$ | | 2.82 ± 0.08 | 40.6 | 84.4 | 2.72 ± 0.04 | 39.2 | 94.4 |
| | biochar | | 2.88 ± 0.09 | 41.5 | 86.3 | 2.36 ± 0.06 | 34.0 | 81.9 |
| Rye | control | 7.50 ± 0.12 | 4.07 ± 0.16 | 54.3 | - | 3.39 ± 0.10 | 45.2 | - |
| | $CaCO_3$ | | 3.74 ± 0.09 | 49.9 | 91.9 | 3.06 ± 0.06 | 40.8 | 90.3 |
| | biochar | | 3.62 ± 0.05 | 48.3 | 89.0 | 2.78 ± 0.14 | 37.1 | 82.1 |
| Oats | control | 7.30 ± 0.10 | 4.62 ± 0.06 | 63.3 | - | 3.65 ± 0.12 | 50.0 | - |
| | $CaCO_3$ | | 4.02 ± 0.10 | 56.4 | 89.1 | 3.30 ± 0.09 | 45.2 | 90.4 |
| | biochar | | 4.00 ± 0.08 | 54.8 | 86.6 | 3.08 ± 0.07 | 42.2 | 84.4 |

± standard deviation, at $p < 0.05$, $n = 4$.

The scheme of the experiment included the following three variations for each plant material:
1. Sand + plant material (control);
2. Sand + plant material + $CaCO_3$ (5%);
3. Sand + plant material + biochar (1%).

*2.3. Laboratory Methods*

The C and N total content in plant materials were determined using an element analyzer (Euro EA3028-HT Analyser, Pavia, Italy). The plant material, ash, was determined by wet oxidation with the Ginzburg method [35]. The total carbon content in compost was determined by the Tyurin dichromate-oxidation method. The total nitrogen content in compost was determined by wet oxidation by the Tyurin microchromic method. The distillation of ammonia was carried out similarly to the Kjeldahl method [3,36]. The enrichment of organic substances with nitrogen was characterized with ratio C/N.

Humic substances determination was based on their properties to dissolve in the alkaline aqueous solutions used as extragents [23,37–39]. HAs and FAs were extracted from compost samples (1:10 compost/solution mass ratio) with 0.1 M NaOH and 0.1 M $Na_4P_2O_7$. Then, HAs were separated from FAs using 0.5 M $H_2SO_4$ solution. The amount of labile HAs (HAs1) was estimated by the content in 0.1 M NaOH-extract, stabilized HAs (HAs2) by the difference between the HAs content in 0.1 M $Na_4P_2O_7$- and 0.1 M NaOH-extracts [24,40].

Preparation of HAs for elemental composition and NMR analysis was performed according to the procedure recommended by the International Humic Substances Society, with minor modifications [38]. HAs were extracted from air-dry compost samples after pretreatment with 0.1 M HCl. Then, they were extracted with 0.1 M NaOH solution (soil solution ratio 1:10). HAs were precipitated with 1 M HCl. To purify the HAs preparations, they were subjected to dialysis [32,41,42].

The $E_C^{mg/mL}$ index was used to assess the maturity of organic matter [26]. This indicator is a quotient of dividing the optical density obtained on a photoelectrocolorimeter by the concentration of carbon in solution expressed in mg per 1 mL. The HAs optical density was measured at 465 nm. The cuvette working length was 1 cm and the reference solution was distilled water [43]. The optical density index reflects the carbon of the humic acids aromatic part to the carbon of their side chains ratio. This index allows to characterize the share of aromatic compounds in HAs molecules [32,44].

The HAs preparations elemental composition was analyzed using the Euro EA3028-HT Analyser. The percentage content of C, N, H, and O, was determined to allow the calculation of atomic ratios O/C, C/H, C/N, and O/H. The oxidation degree of HAs molecules was calculated by the formula: $\omega = (2Q_O - Q_H)/Q_C$, where $Q_O$ is the number of O atoms, $Q_H$ is the number of H atoms, and $Q_C$ is the number of C atoms.

Solid-state $^{13}$C-NMR HAs spectra were measured using the Bruker Avance 500 NMR spectrometer by a 3.2 mm $ZrO_2$ rotor (the research was carried out on the basis of the St. Peters). Solid-phase samples (fine grounder powder) were placed in a 4-mm zirconium oxide rotor and spun at a frequency of 12.5 kHz at the magic angle. The cross-polarization sequence of excitation pulses was used for the registration of $^{13}$C-spectra Agronomy 28.08.2022-12(9)2053 (CP/MAS). The contact time was 2 m, the delay time was 2 s, and the number of scans was 8000. Chemical shifts were referenced against tetramethylsilane. Relative contributions of the various carbon groups were determined by integration of the signal intensity in their respective chemical shift regions (Saint-Petersburg State University "Magnetic Resonance Research Methods" Resource Center). NMR spectra were decoded using the Mestrenova software with correction of the baseline spectra. Spectras were deciphered on the base of chemical shift values; every interval of chemical shift is characteristic for a particular group of chemical compounds. The following ranges and corresponding groups of structural carbon fragments were distinguished: C, H-linked aliphatic groups (0–47 ppm), Methoxyl and O, N-linked aliphatic fragments (47–60 ppm), Aliphatic fragments, twice substituted heteroatoms (including carbs) and esters, and ethers (60–110 ppm), C, H-linked aromatic fragments; O, N-linked aromatic fragments (110–160 ppm), Carboxyl groups, esters, amides and their derivatives (160–185 ppm), and Quinone groups; and Aldehyde and ketone groups (185–200 ppm). The total content of aromatic and aliphatic groups and their Ar/Al ratio were also calculated [45,46].

*2.4. Statistical Analyses*

The experiment consisted of four independent measurements of the parameter investigated: four compost samples were taken from four pots. Data were subjected to analysis of variance procedures (one-way ANOVA). The statistical significances of differences between the mean values were determined by Student-Newman-Keuls test at $p < 0.05$. Equality of variances was evaluated using the Levene test. Statistical data processing was performed using the IBM SPSS Statistics, Version 25(«IBM», New York, NY, USA).

## 3. Results and Discussion

### 3.1. Mineralization

The addition of calcium carbonate and biochar to the composted mass changed the dynamics of the transformation processes of plant residues. Regardless of the enrichment of plant material with nitrogen under the influence of the additives, the activation of mineralization of organic substances was observed (Table 2).

At the same time, this effect appeared not only in the initial stage of composting, when the plant material is still rich in compounds more accessible for biodegradation, but it also remained until the end of the experiment. Evidently, both calcium carbonate and biochar intensified the decomposition of less immobilized organic compounds. The greatest effect during the all the experiment was observed with the joint composting of biochar and plant residues, regardless of their chemical composition. The mineralization of rye and clover residues was by 18%, and oats by 16%, more than in the control variant, while the addition of calcium carbonate in rye and oat compost increased mineralization by 10%, and in clover compost, by 6%.

Considering the possible mechanism of the observed effect, it can be assumed that the intensification of the plant material mineralization processes during composting, together with biochar, occurs due to the introduction of an additional source of mineral nutrition for microorganisms into the system. However, the green mass of clover is rich in easily degradable compounds of both carbon and nitrogen. Nevertheless, even during the decomposition of this plant material, the stimulation of the mineralization process was observed. Therefore, without excluding the possible influence of biochar as a material source of nutrition and energy for microorganisms, it should also be assumed that there is another as yet unexplored mechanism for activating this process.

*3.2. Humification*

Before assessing the peculiarities of the humification processes, it should be noted that no HAs were detected in the initial plant materials (before composting) (Table 3).

**Table 3.** Quantity and optical properties of plant residue components soluble in 0.1 M NaOH-extract.

| Variants | | The Sum of Soluble Substances | | Sum of Substances Precipitated by 0.5 M of $H_2SO_4$ | |
|---|---|---|---|---|---|
| | | % on $C_{tot}$ | $E_C$ mg/mL | % on $C_{tot}$ | $E_C$ mg/mL |
| Clover | control | $2.1 \pm 0.1$ | $0.8 \pm 0.1$ | $2.1 \pm 0.1$ | $0.8 \pm 0.1$ |
| | $CaCO_3$ | $1.6 \pm 0.1$ | $0.5 \pm 0.0$ | $1.3 \pm 0.2$ | $0.5 \pm 0.1$ |
| | biochar | $2.0 \pm 0.2$ | $0.5 \pm 0.1$ | $2.0 \pm 0.1$ | $0.6 \pm 0.1$ |
| Rye | control | $1.5 \pm 0.1$ | $0.8 \pm 0.1$ | $1.2 \pm 0.2$ | $0.8 \pm 0.1$ |
| | $CaCO_3$ | $1.8 \pm 0.2$ | $0.7 \pm 0.1$ | $1.1 \pm 0.2$ | $0.7 \pm 0.1$ |
| | biochar | $1.4 \pm 0.1$ | $0.7 \pm 0.1$ | $1.2 \pm 0.3$ | $0.8 \pm 0.1$ |
| Oats | control | $2.0 \pm 0.1$ | $0.4 \pm 0.1$ | $1.4 \pm 0.1$ | $0.5 \pm 0.1$ |
| | $CaCO_3$ | $1.6 \pm 0.1$ | $0.4 \pm 0.0$ | $1.3 \pm 0.1$ | $0.4 \pm 0.1$ |
| | biochar | $1.7 \pm 0.1$ | $0.5 \pm 0.0$ | $1.5 \pm 0.1$ | $0.6 \pm 0.1$ |

$\pm$ standard deviation, at $p < 0.05$, $n = 4$.

This is evidenced by the insignificant amount of organic compounds soluble in 0.1 M NaOH, and the fraction precipitated by acid, as well as their very low optical density index, which is uncharacteristic for HAs [26,43].

The additives introduced significantly changed the nature and volumes of HAs newly formed, as well as their further transformation (Figure 1).

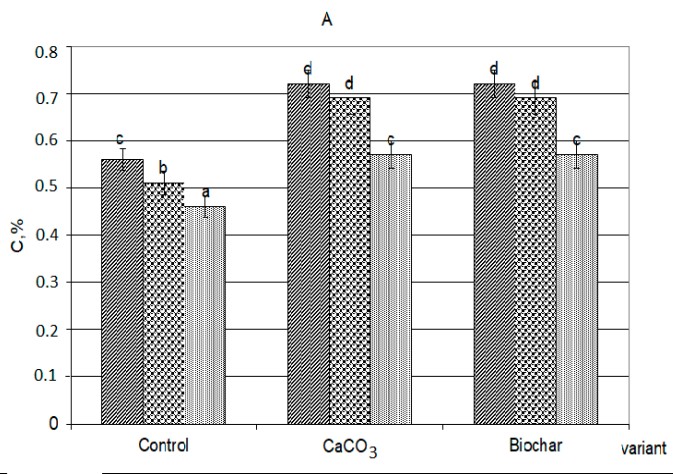 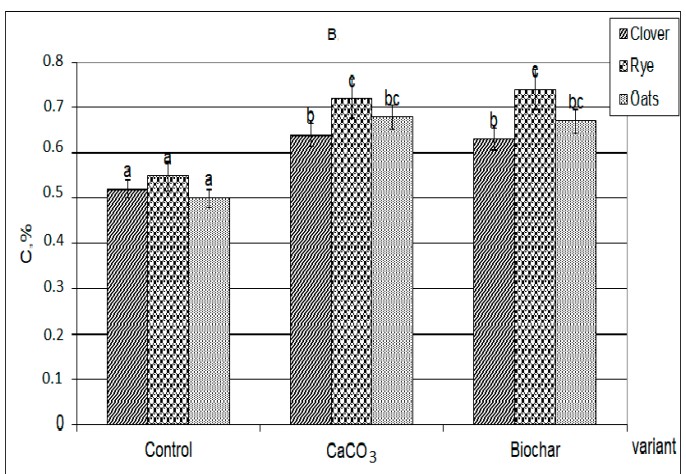

**Figure 1.** The amount of newly formed humic acids formed during the decomposition of plant residues extracted with 0.1 M $Na_4P_2O_7$ solution, C, %: (**A**)—30 days, (**B**)—90 days (the same small letters mean no significant differences between the values at the level of significance $p < 0.05$, $n = 4$; one-way ANOVA, Student-Newman-Keuls test).

First, in the presence of calcium carbonate and biochar, the HAs formation was significantly more active than in the control variants. Against the background of additives, more organic substances were involved in humification (by 20–30%). The composts with additives practically did not differ from each other by the total amount of formed HAs (extracted by sodium pyrophosphate solution from the decomposing plant residues). Secondly, during the decomposition of plant residues together with both $CaCO_3$ and biochar, the formation of calcium humates (Figure 2), forms of HAs that are highly resistant to the

decomposing action of microorganisms and completely uncharacteristic for young weakly humified compounds, were observed [24,27,28].

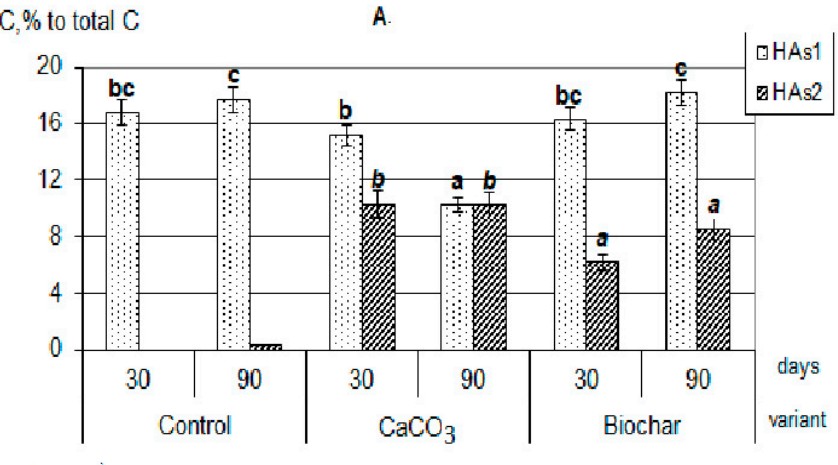

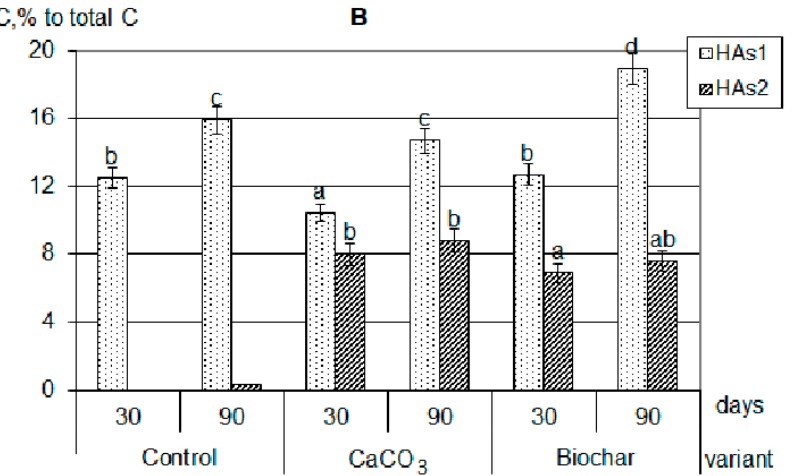

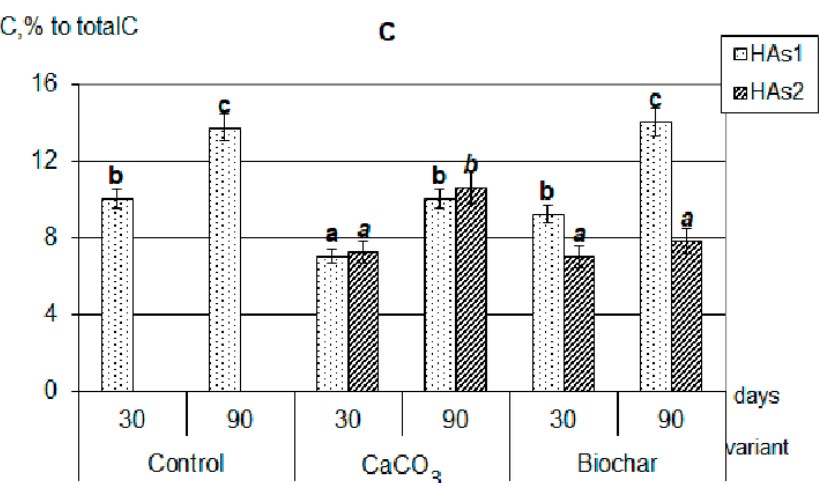

**Figure 2.** Composition of newly formed humic acids formed during the decomposition of plant residues, C, % to total C: (**A**)—clover, (**B**)—rye, (**C**)—oats (the same small letters mean no significant differences between the values at the level of significance $p < 0.05$, $n = 4$; one-way ANOVA, Student-Newman-Keuls test).

Significant differences in the effect of the studied reagents were evident in the assessment of the structure and properties of the newly formed HAs, as evidenced by the results of their optical density and elemental composition (Figure 3, Table 4). It should be noted that all compounds formed during the decomposition of plant residues, despite the low value of the optical density index, can be reasonably classified as brown HAs [28,41].

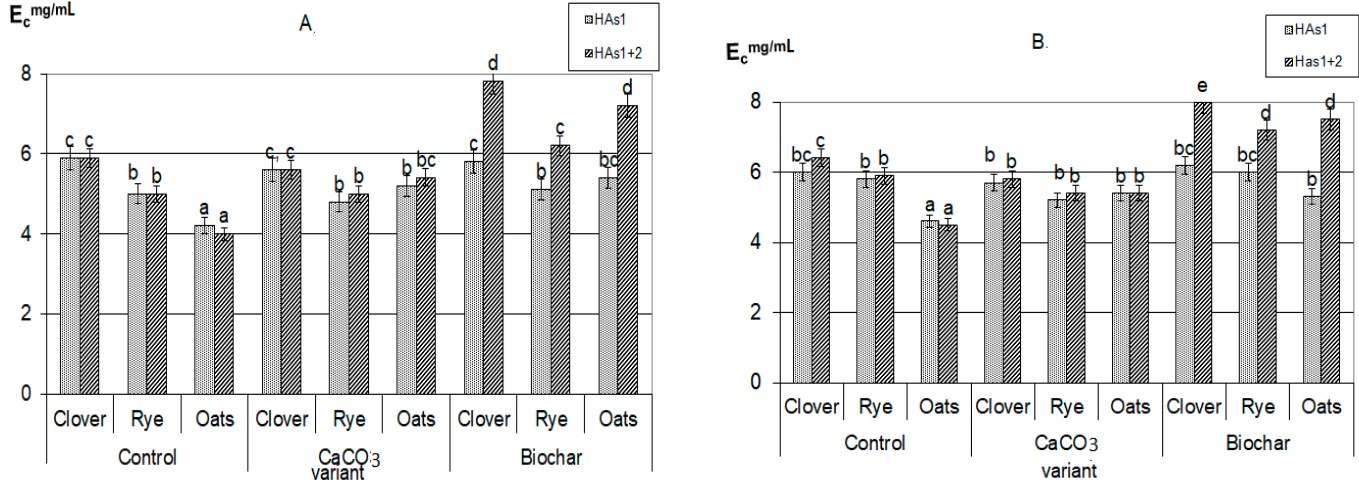

**Figure 3.** The index of optical density of humic acids formed during plant residues decomposition: (**A**)—30 days, (**B**)—90 days (the same small letters mean no significant differences between the values at the level of significance $p < 0.05$, $n = 4$; one-way ANOVA, Student-Newman-Keuls test).

**Table 4.** The elemental composition, atomic ratio, and oxidation degree of humic acids (HAs1 + HAs2) and initial biochar.

| Variant | | C | N | H | O | H/C | O/C | O/H | C/N | ω |
|---|---|---|---|---|---|---|---|---|---|---|
| | | \multicolumn Content, at % | | | | | | | | |
| Clover | control | 37.31 | 2.69 | 41.88 | 18.12 | 1.12 | 0.49 | 0.43 | 13.87 | −0.15 |
| | CaCO$_3$ | 36.66 | 2.51 | 42.17 | 18.65 | 1.15 | 0.51 | 0.44 | 14.60 | −0.13 |
| | biochar | 38.09 | 2.39 | 41.22 | 18.30 | 1.08 | 0.48 | 0.44 | 15.94 | −0.12 |
| Rye | control | 34.99 | 1.60 | 46.14 | 17.26 | 1.32 | 0.49 | 0.37 | 21.87 | −0.33 |
| | CaCO$_3$ | 35.64 | 1.71 | 45.19 | 17.47 | 1.27 | 0.49 | 0.39 | 20.84 | −0.29 |
| | biochar | 37.23 | 1.84 | 43.23 | 17.69 | 1.16 | 0.48 | 0.41 | 20.23 | −0.21 |
| Oats | control | 37.54 | 1.62 | 45.66 | 17.26 | 1.22 | 0.46 | 0.38 | 23.17 | −0.30 |
| | CaCO$_3$ | 36.08 | 1.56 | 44.95 | 17.42 | 1.24 | 0.48 | 0.39 | 23.13 | −0.28 |
| | biochar | 36.26 | 1.60 | 44.43 | 17.74 | 1.22 | 0.49 | 0.40 | 22.66 | −0.25 |
| Origin biochar | | 67.60 | 0.30 | 35.10 | 3.40 | 0.52 | 0.05 | 0.10 | 225.33 | −0.42 |

It should be emphasized that in all composts with added calcium carbonate, the optical density index of HAs extracted with sodium pyrophosphate (HAs1 + HAs2) practically did not change during the whole composting period, and was significantly lower than in composts with added biochar (Figure 3). In addition, there were no differences in the values of optical density HAs1 + HAs2 when compared with labile forms of HAs1. Thus, the appearance of stable forms in variants with the introduction of CaCO$_3$ did not lead to a deepening of the humification processes, that is, it did not fundamentally change their nature. This allowed the probable mechanism of calcium humates formation under the influence of CaCO$_3$ to consider the regrouping of some fractions similar to the processes occurring during soil liming [30]. With the process of HAs regrouping under soil liming,

the part of HAs1 transit to HAs2, however, such conversion does not practically affect the properties of the resulting fraction. Such a state will be maintained only with a significant excess of Ca in the system, and with the decrease of its content the reverse regrouping will be observed.

On the contrary, in the variants with the addition of biochar, the humification process actively continued throughout the entire composting time. In all compost with a biochar, the sum of HAs (HAs1 + HAs2) optical density was significantly higher than of only labile forms (HAs1) and increased during the experiment. The maximum value of the HAs optical density index was reached in composts with clover (Figure 3); all this indicates a higher degree of HAs aromaticity, an increase in their chemical "maturity", and a deepening of the humification process during the composting period. This generally increases the resistance of newly formed HAs to microbiological and biochemical actions. Probably, the mechanism of formation of stabilized HAs is associated both with the rearrangement of fractions (since Ca is also present in large quantities in the composition of biochar) and with the positive biochar influence on microbiological activity in the compostable material.

Discussing the mechanism of stable HAs fractions formation at the early stages of humification, it is impossible to exclude sorption of newly formed compounds directly by biochar, which can increase their resistance to extraction. In addition, the role of biochar as a material source of HAs formation should not be ruled out when biochar is included in the humic complex. However further careful research is required to confirm these assumptions.

### 3.3. Elemental Composition and $^{13}$C-NMR Data of HAs

In addition to optical density characteristics, the elemental composition analysis was used to assess the quality of newly formed compounds. For this, at the final stage of the experiment (90 days), HAs preparations were isolated from all the obtained composts. To obtain HAs (HAs1 + HAs2) the extraction was performed with 0.1 M NaOH after pretreatment with 0.1 M HCl. The results of the elemental composition of the studied HAs are presented in Table 4.

The elemental composition of all the studied preparations fully corresponded to the parameters of the HAs group, regardless of the plant material type and the added reagent [23,24]. Their composition is specific for newly formed HAs with a reduced carbon and oxygen content, as well as a high H/C ratio [5,7,47]. Such compounds are characterized by a benzoidicity low degree, an insignificant content of acidic functional groups, and as a consequence a lack of affinity for calcium. Therefore, the presence of calcium humates in their composition is not obvious [26,48].

The analysis of the results obtained in this experiment showed that both plant residues and the tested reagents influenced the formation of the HAs elemental composition.

The influence of the plant residues composition on the HAs elemental composition in all experiment variants was manifested very clearly. In the HAs elemental composition the maximum differences were noted in the nitrogen content, which increased in clover compost and decreased in cereal compost. The clover plant residue compost was characterized by the highest HAs humification of all those studied. This is evidenced by a relatively high content of C, N, and O, and a lower content of H, as well as the increased atomic ratio of O/H and the reduced ratios of H/C and C/N. With that, according to the considered results, the HAs formed in all three variants of clover compost (control, with calcium carbonate, and with biochar) not only exceed the HAs of rye and oat compost but are quite comparable with soil HAs [24]. Thus, the high nitrogen enrichment of the initial plant residues was of great importance for all stages of humification (mineralization and humus formation), and contributed to a more active and efficient flow of this process.

Evaluating the results of the studied reagents' influence on the HAs elemental composition, it should be noted that the addition of calcium carbonate to compost did not provide a noticeably positive effect. More significant changes were observed under the influence of biochar. In the newly formed HAs obtained in the presence of biochar, an increase in C, but a decrease in the H proportion in the molecules, was observed. Additionally, the

active influence of the biochar on the HAs formation has been clearly manifested when determining the oxidation degree, which is one of the most informative indicators characterizing the HAs elemental composition [24]. In general, the process of humification is characterized by an increase in the oxidation degree of the formed compounds. According to this indicator, it is possible to assess the intensity and direction of the process. The increase in this indicator was observed in this experiment in all three compost variants under the influence of both calcium carbonate and biochar. Consequently, with the joint composting of all three plant residues types and reagents, the humification processes are taking place more actively that in the control variants. Apparently, the noted changes in the preparations' elemental composition are caused by the replenishment of HAs with more humified and relatively chemically mature compounds.

The data obtained may indicate a partial restructuring of the HAs molecules' carbon skeleton, a decrease in the proportion of aliphatic chains as an effect of their partial destruction, an increase in the aromatization degree, and in the oxygen-containing functional groups content. This is consistent with the study of the HAs optical density results (Figure 3). In general, the noted changes indicate an increase in the stabilization and stability of the HAs. The maximum positive changes in the HAs elemental composition were revealed during the joint humification of clover and biochar.

It should also be noted that, for all the studied parameters of the elemental composition, biochar significantly differs from the HAs (Table 4).

To obtain additional information about changes in the molecular structure of the formed under the influence of reagents, HAs were selected for analysis preparations isolated from the compost of clover mass. Additionally, with the aim of evaluating if it had a possible direct participation in the formation of Has, the initial biochar was stated. There samples were investigated using the $^{13}$C-NMR spectroscopy method (Figure 4, Table 5).

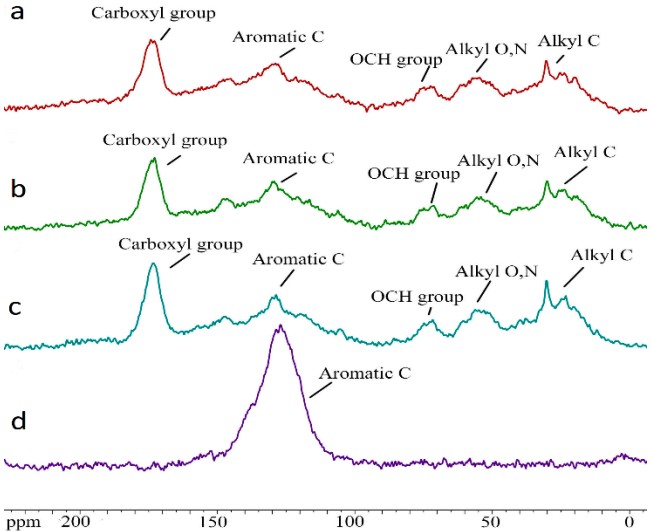

**Figure 4.** CPMAS $^{13}$C-NMR spectra HAs humic acids formed during the decomposition of plant residues of clover, 90 days of composting ((**a**)—control; (**b**)—CaCO$_3$; (**c**)—biochar; (**d**)—initial biochar).

The obtained data are characteristic of the HAs molecular composition [46,49,50]. The content of aromatic fragments in the newly formed HAs is significantly lower than in the initial biochar, but it still remains quite high (Table 5). The organic matter decomposition degree varies in a wide range. The content of quinone, aldehyde, and ketone groups, is approximately comparable in all variants, which means that this part of the HAs is not affected by the experiment. At the same time, the carboxyl groups content is minimal in the variant, with the addition of biochar.

**Table 5.** Contents of molecular fragments of HAs formed during decomposition of clover leaves, 90 days of composting (from $^{13}$C-NMR data).

| Variant | Chemical Shift, ppm | | | | | | Aliphatic Group | Aromatic Group | AR/AL |
|---|---|---|---|---|---|---|---|---|---|
| | 0–47 | 47–60 | 60–110 | 110–160 | 160–185 | 185–200 | | | |
| Control | 32 | 7 | 3 | 27 | 24 | 7 | 49 | 51 | 1.04 |
| CaCO$_3$ | 36 | 8 | 2 | 24 | 24 | 6 | 52 | 48 | 0.92 |
| Biochar | 25 | 7 | 18 | 30 | 16 | 4 | 54 | 46 | 0.85 |
| Origin biochar | 15 | 4 | 16 | 55 | 6 | 4 | 39 | 61 | 1.56 |

Furthermore, the content of aliphatic groups, double-substituted heteroatoms (including carbohydrates), and esters and ethers, is increased in the variant with biochar. However, in general, there were no significant differences between HAs in the results of compost $^{13}$C-NMR analysis, which apparently is associated with the active replenishment of HAs by forming labile compounds. For a deeper characterization of newly formed Has, further separation of them, according to the resistance to microbiological effects degree, is required (separation of fractions 1 and 2).

As for the comparative analysis of the composition and molecular structure of the HAs obtained in the process of composting, and the original biochar used for application to plant residues, the data obtained indicate their completely different nature (Tables 4 and 5). The elemental composition of the original biochar differed significantly from the composition of the newly formed HAs. It was characterized by very high carbon content, extremely low oxygen content, and practical absence of nitrogen. The high content of polycyclic aromatic rings in the original biochar, the increased degree of aromaticity of the material, and the extremely low degree of the organic matter decomposition, also indicate a fundamentally different molecular structure of HAs and biochar. This does not give argument to consider the biochar as a material source of the HAs formation.

## 4. Conclusions

The composting of plant residues characterized by different nitrogen enrichment (fresh, aboveground mass of clover and rye, well as dry oat straws), together with calcium carbonate and biochar, were studied. The research results showed that both reagents had a positive effect on composting processes. The additives significantly changed the nature and the volume of newly formed HAs and influenced their further transformation. The HAs formation proceeded much more actively—20–30% more HAs were formed with calcium carbonate and biochar than in the control variants. For the first time, the transformation of newly formed HAs in stable forms (calcium humates), which is completely uncharacteristic for "young" slightly humified compounds, was shown. The proportion of this form in the whole HAs composition constitutes 40–50%.

However, the mechanism of HAs transformation under the influence of the studied reagents was different. In the presence of calcium carbonate, it is caused by the physicochemical processes of the newly formed HAs rearrangement, without changing their chemical maturity. In the presence of biochar, this mechanism is due to the humification processes' intensification, and due to the increase in the aromatization degree (the increase in the proportion of carbon and oxygen, and the decrease in hydrogen in HAs molecules).

As for the biochar role in the humus formation processes, the different molecular structure of Has and biochar was shown. Therefore, there is no reason to consider biochar as a material source of HAs formation at this research stage. Its role is in the ability to activate humic substance formation, and to increase the proportion of Has, including their stable forms. This leads to an increase of the newly formed humic substances system stabilization as a whole. Understanding of HAs formation and transformation mechanisms during composting of organic residues can help to optimize the methods of obtaining and applying organic fertilizers in order to improve the quality and stability of soil organic

matter, to preserve and reproduce fertility, and generally to have a positive impact on the agroecological state of ecosystems.

**Author Contributions:** Conceptualization, obtaining funding by N.O. and E.O.; incubation experiments, research, E.O. and K.S.; writing—preparation of the initial draft, N.O.; writing—reviewing and editing, N.O., E.O. and E.A.; 13C NMR analysis, E.A.; elemental analysis and review by S.C. All authors have read and agreed to the published version of the manuscript.

**Funding:** This study was supported by the Russian Science Foundation (Grant number: 22-26-00187).

**Data Availability Statement:** Center of Chemical Analyses and Materials and Center of Magnetic Resonance Research, Scientific Park of Saint-Petersburg State University. The data of NMR spectroscopy has been obtained from the "Center of Chemical Analyses and Materials and Center of Magnetic Resonance Research".

**Acknowledgments:** Authors thanks Research Park of Saint-Petersburg State University.

**Conflicts of Interest:** The authors declare no conflict of interest.

## Abbreviations

| | |
|---|---|
| HAs | humic acids |
| FAs | fulvic acids |
| HAs1 | labile HAs |
| HAs2 | stabilized HAs |
| NMR | nuclear magnetic resonance |

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
