# Peer review of "Humic Acids Formation during Compositing of Plant Remnants in Presence of Calcium Carbonate and Biochar"

_agronomy, doi:10.3390/agronomy12102275_

Round 1

Reviewer 1 Report (Previous Reviewer 2)

1.  Proofreading error in the abstract   550oC

2. Title of Table 1 "Characteristics of plant material" is not correct. There are not characteristics, there is concentration of some plant materials and ash.

3. The function of the letters a, b, c, d in the figures can’t be understood, please explane.

4. Obscure abbreviations; s HAs1, FA1, HAs2, FA2, CPMAS 13C-NMR, EA3028-HT, 3C-NMR. Only the abbreviation humic acids is explained (HAs)

5. Proofreading error in the Table 2       82,1

6. In all figures, the indices of the formulas (CaCO3) are written incorrectly

7. In Figures 1, 2 and 3, the font is unequal and the wrong size.

The content has been revised, but there are many flaws in layout and proofreading.

Author Response

Manuscript ID: agronomy-1813091. Type of manuscript: Article. Title: Humic acids formation during compositing of plant remnants in presence of calcium carbonate and biochar. Authors: Nataliya Orlova *, Elena Orlova, Kseniia Smirnova, Evgeny Abakumov, Serafim N. Chukov

Response to Reviewer 1.

Dear reviewer!

We express our deep appreciation for analysis of our manuscript, for your recommendations and comments.

We agree with the all remarks and tried to take them into account and correct our text.

 1. In the Annotation 550oC – corrected.2. The name of Table 1was changed to "Concentration of C, N and ash in plant material".3. Fig. 1, 2, 3 – "one-way ANOVA, Student-Newman-Keuls test" was added to the caption.4. "Euro EA3028-HT Analyser" was added to the Methods.5. In Table 2 CaCO3 – corrected; in "82.1" the comma was corrected.6. All (HAs, FAs, EA 3028-HT, NMR) abbreviations have been deciphered in the text and the list of abbreviations was added to the text.  

Thank you for your consideration.

Sincerely,

Nataliya Orlova

Dr. Science

Dept. of Agriculture, Faculty of Biology, Saint-Petersburg State University, 199178, 16 line V.O., 29, Saint-Petersburg, Russia, norlova48@mail.ru, tel. +79219281664

Reviewer 2 Report (Previous Reviewer 1)

The manuscript has been greatly improved.

Curve d in Figure 4 is incomplete.

Author Response

Manuscript ID: agronomy-1813091. Type of manuscript: Article. Title: Humic acids formation during compositing of plant remnants in presence of calcium carbonate and biochar. Authors: Nataliya Orlova *, Elena Orlova, Kseniia Smirnova, Evgeny Abakumov, Serafim N. Chukov

Response to Reviewer 2.

Dear reviewer!

 We express our deep appreciation for analysis of our manuscript and your recommendations.We would like to provide clarification for your comments. It is clarified now that curve d in Figure 4 is 13C-NMR spectra of biochar.   

Thank you for your consideration.

Sincerely,

Nataliya Orlova

Dr. Science

Dept. of Agriculture, Faculty of Biology, Saint-Petersburg State University, 199178, 16 line V.O., 29, Saint-Petersburg, Russia, norlova48@mail.ru, tel. +79219281664

Reviewer 3 Report (New Reviewer)

The authors present an interesting paper looking at the influence of calcium carbonate and biochar on the humification of composing plants with varying levels or organic N content.The results the authors report support their final conclusion that these additives do play a role in the process.

This paper needs extensive English language editing to improve comprehension and contains typos, gramatical errors, and passages with missing words too numerous to list here. These issues have impacted by ability as a reviewer to fully connect specific observations to conclusions in the paper. These errors are most significant in the passages highlighted in green.

More details are needed to desire the methods for the NMR measurements (spinning speed, cross polarization contact time, number of scans).

Spectra D in figure4 is cut off at the bottom.

Line 13: Sentence is not clear - word missing?
Line 18: What is meant by "the obtained"? Is there a word missing?

Line 21: "Mixture"

Line 26" Thire" ?

Line 95: "Primarily with Ca and in the f this form..." Word missing?

Line 245: Does "more hardly immobilized" mean "less mobilized"?

Table 5 - "Alifatic" should be "aliphatic", "groop" should be "group"

Author Response

Manuscript ID: agronomy-1813091. Type of manuscript: Article. Title: Humic acids formation during compositing of plant remnants in presence of calcium carbonate and biochar. Authors: Nataliya Orlova *, Elena Orlova, Kseniia Smirnova, Evgeny Abakumov, Serafim N. Chukov

Response to Reviewer 3.

Dear reviewer!

 We express our deep appreciation for analysis of our manuscript and your recommendations.

We agree with the remarks and tried to take them into account and correct our text.

We would like to provide clarification for your comments. 1. More detailed description of the NMR measurements was added to the Methods.2. Line 21 – corrected a typo.3. Line 26 – corrected a typo.4. Line 95 – removed the typo.5. Line 245 – corrected to "less mobilized".6. In Table 5 – corrected typos.   

Thank you for your consideration.

Sincerely,

Nataliya Orlova

Dr. Science

Dept. of Agriculture, Faculty of Biology, Saint-Petersburg State University, 199178, 16 line V.O., 29, Saint-Petersburg, Russia, norlova48@mail.ru, tel. +79219281664

Round 2

Reviewer 3 Report (New Reviewer)

This paper still needs English language editing to improve clarity. The methods used are appropriate and the conclusions are supported by the results.  

Corrections for the revised text:

Line 195: what is meant by “Agronomy 2022, 12, 2053 5 of 21” ?

Line 196: should read: “contact time was 2 ms”, I assume.

This manuscript is a resubmission of an earlier submission. The following is a list of the peer review reports and author responses from that submission.

Round 1

Reviewer 1 Report

Dear authors, overall, research of the different organic fertilizers based on compost and substrates in order to apply it in landscaping and agricultural sectors are timely and necessary due to intensive soil degradation around the world. However, your study doesn't have a universal concept and focuses only on three types of plant residues. I was not able to find the convincing novelty, scientific hypotheses, and even recommendations for practical application in a broader sense. In this work You have presented only the description of the chemical properties of the samples without a fruitful discussion of the obtained results and the possible implementations in a broader perspective.

Many cited references are outdated or of marginal importance. In particular, references 20, 25 and 26 are from marginal journals, e. g. "Bulletin of Moscow University. 17, Soil science" and similar. The rest of the references are also of local origin as well as difficult to access, such as older Russian works (references 19, 31-41). For the work to be relevant in modern days, references need to reflect the current state of the art. A cursory look at the literature suggests there are more relevant, up to date references that could be used instead.

The Section Materials and methods is well structured. Authors should clarify the information about the ashes in Table 1 in the subsection 2.1 Characteristics of plant material and biochar. In the subsection 2.2 Incubation experiment, the authors should specify the description of the experimental scheme and the reasons for the selection of 5% CaCO3 or 1% biochar additive. It is not clear what is biochar (1) (Line 119). In subsection 2.3 Laboratory methods the nitrogen determination method is incorrectly specified (line 122-124) and cited reference (26) is also incorrect. It is necessary to explain what E4 is (Line 133). The methodology for determining optical density index Ec, mg/ml is presented inconsistently and unclearly. Literature references are not available (25), or not very suitable, cross-referenced (29). Only 30th literature source (Line 134) is suitable to support the methodology.

Authors should provide sufficient detail tables and figures in the Section Results and discussion. For example, in Table 2 it is not clear, what the authors wanted to say with this data: initial content or content original.

As a recommendation for presenting the results of this study, the authors must clarify and unify the names and terms in the entire description of the experiment, and it is also necessary to harmonize the results in the text and in the tables and figures. For example, the results presented in the comments (Line164-166) do not match the data in Table 2.

It is not clear, what the authors wanted to say in Figure 1 (Line 230-232): a, b, c (different letters denote the average values that differ from each other at P < 0.05, n = 4). What are the values? The same is true for Figure 2.

The Figure 4 shows unclear numbering of the curves (5-9), but where are 1-4? There are also formatting problems in Table 3 (line 205-216) and Table 4 (line 313-330 and 340-342).

Figure 3 has confusing/incorrect units of measurement.

In the entire paper, as well as in Table 4, the experimental results do not really show a very significant change due to the influence of biochar (Line 353-356). The discussion of the final results does not evaluate and discuss the different carbon content in the starting material.

The abbreviation BC appears in the conclusions, it was not included in the entire paper before that and is not explained anywhere (Line 411.) Conclusions are rather succinct and provide little information. I believe they could be expanded by including some of the results in greater detail.

Reviewer 2 Report

Abstract is not concrete.  It’s Impossible to understand abbreviations in the lines 24-25.
 There is a lot of recurrence in the introduction (lines 47-51, 64-65, 81-83).
Assignments who are written after the objective look like school work.
The comments are not coordinated with the data presented in the tables and graphs:
-          -The plant material was characterized by different nitrogen enrichment (Table 1).  
Is only nitrogen important? Are other elements useless? In this case, why they were determined?

  Plant residues with or without additives were mixed with calcined at 700 °C quartz sand and incubated at a temperature of 25°C and optimal compost humidity (60 % of total water holding capacity) for 90 days.
What kind of plants residues? This is very important because the amount of nutrients in the roots, stems, leaves, and flowers is different.
Results are given after 30 and 90 days.

 -          The total nitrogen content was determined by the Tyurin microchrome method.
Total carbon is determined by Turin's method

 -Regardless of the enrichment of plant material with nitrogen under the influence of the additives, the activation of mineralization of OM was observed (Table 2). The mineralization of rye and clover residues was by 18% and oats by  16% more than in the control variant while the addition of calcium carbonate in rye and  oat compost increased mineralization by 10% and in clover compost – by 6%.
There is no such data.

 -The sentences contradict each other:

“Thus, the appearance of  stable forms in the system of newly formed HAs did not lead to the deepening of humification processes, i.e., it did not change their nature in principle.”
and
“All this indicates a higher degree of HAs aromaticity, an increase in their chemical "maturity" and in general a deepening of the humification process during the composting period.”

 -          In contrast, in the versions with the addition of biochar, the process of humification actively continued throughout the duration of composting.
There is no difference, same dependency as in other cases.

 The function of the letters a, b, c, d in the figures can’t be understood.
Incorrect figures imaging.
The index in the formula (CaCO3) is written incorrectly, there are Russian inscriptions in Fig. 3, the abbreviation BC appears from somewhere, and many other proofreading errors.
References without DOI. There is no way to verify the appropriateness and accuracy of the citation.

In the article there are many unclear and unreasonable sentences, they are marked in colours.

Extra careful revision and editing of the article is required.

Reviewer 3 Report

Analyzes of soil organic matter, including humic substances, constitute an important research aspect in the processes of carbon sequestration in soil. Therefore the submitted paper entitled ‘Humic acids formation during compositing of plant remnants in presence of calcium carbonate and biochar’ seems interesting, but nevertheless contains many methodological errors and inadequate terms.

Firstly, The authors say that they are investigating the humification and mineralization ‘processes’ while investigating the specific ‘state’ of the material by NMR after 90 days of incubation. In addition, both calcium carbonate and biochar may not be treated as the course indicators of the observed changes in the vegetable material due to completely different physicochemical properties and availability to mineralization.

Another issue concerns the method of humic acids extractions. The Authors chose the extraction of only 0.1M NaOH without the previously performed HCL extraction, which means that they also studied fulvic acids, which are formed in the first stage of humus formation and are the most sensitive to the processes of changing environmental conditions. This makes the results unreliable. I suggest the Authors to read the latest literature and data included on the International Humic Substances Society website because the Bielczikowa-Kononowa method is now outdated.

Moreover, some statements in the text are incorrect, such as ”newly formed labile humic substances in forms relatively resistant to biochemical influences” - this is not true because low-molecular fulvic acids, which are labile forms, are largely susceptible to depletion of biochemical processes due to their low molecular weight.  There are many similar statements in this publication, therefore I suggest to reject the paper.